# Antioxidant Phytocomplexes Extracted from Pomegranate (*Punica granatum* L.) Using Hydrodynamic Cavitation Show Potential Anticancer Activity In Vitro

**DOI:** 10.3390/antiox12081560

**Published:** 2023-08-04

**Authors:** Antonella Minutolo, Angelo Gismondi, Rossella Chirico, Gabriele Di Marco, Vita Petrone, Marialaura Fanelli, Alessia D’Agostino, Antonella Canini, Sandro Grelli, Lorenzo Albanese, Mauro Centritto, Federica Zabini, Claudia Matteucci, Francesco Meneguzzo

**Affiliations:** 1Department of Experimental Medicine, University of Rome Tor Vergata, 00133 Rome, Italy; antonella.minutolo@uniroma2.it (A.M.); chirico.rossella3@gmail.com (R.C.); vita.petrone01@gmail.com (V.P.); fanellimarialaura@gmail.com (M.F.); grelli@med.uniroma2.it (S.G.); matteucci@med.uniroma2.it (C.M.); 2Department of Biology, University of Rome ‘Tor Vergata’, Via della Ricerca Scientifica 1, 00133 Rome, Italy; gismondi@scienze.uniroma2.it (A.G.); gabriele.di.marco@uniroma2.it (G.D.M.); d.agostino@scienze.uniroma2.it (A.D.); canini@uniroma2.it (A.C.); 3Virology Unit, Policlinic of Tor Vergata, 00133 Rome, Italy; 4Institute of Bioeconomy, National Research Council of Italy, Via Madonna del Piano 10, 50019 Florence, Italy; lorenzo.albanese@cnr.it (L.A.); federica.zabini@cnr.it (F.Z.); 5Institute for Sustainable Plant Protection, National Research Council of Italy, Via Madonna del Piano 10, 50019 Florence, Italy; mauro.centritto@cnr.it

**Keywords:** anti-cancer, antiradicals, bioactive metabolites, green extraction, hydrodynamic cavitation, phytochemicals, pomegranate

## Abstract

Hydrodynamic cavitation (HC), as an effective, efficient, and scalable extraction technique for natural products, could enable the affordable production of valuable antioxidant extracts from plant resources. For the first time, whole pomegranate (*Punica granatum* L.) fruits, rich in bioactive phytochemicals endowed with anti-cancer properties, were extracted in water using HC. Aqueous fractions sequentially collected during the process (M1–M5) were lyophilized (L), filtered (A), or used as such, i.e., crude (C), and analyzed for their biochemical profile and in vitro antioxidant power. The fractions M3 and M4 from the L and C series showed the highest antiradical activity and phytochemical content. While the lyophilized form is preferable for application purposes, sample L-M3, which was produced faster and with lower energy consumption than M4, was used to assess the potential antiproliferative effect on human breast cancer line (AU565-PAR) and peripheral blood mononuclear (PBMC) cells from healthy donors. In a pilot study, cell growth, death, and redox state were assessed, showing that L-M3 significantly reduced tumor cell proliferation and intracellular oxygen reactive species. No effect on PBMCs was detected. Thus, the antioxidant phytocomplex extracted from pomegranate quickly (15 min), at room temperature (30 °C), and efficiently showed potential anticancer activity without harming healthy cells.

## 1. Introduction

According to the World Health Organization (WHO), cancer is one of the leading causes of death worldwide. In terms of prevention, the focus is on interventions to lower the incidence of the disease by reducing exposure to risk factors and attempting to increase the health resistance at the individual level. To date, the most commonly used treatments are chemotherapy, radiotherapy, hormone therapy, immunotherapy, surgery, etc., either as a single strategy or possibly in combination [1]. Standard protocols, however, have several downsides, including not being able to distinguish between healthy and cancerous cells. In fact, the main expected features of any antitumor treatment should be the lack of toxic effects and the ability to kill tumor cells without damaging healthy tissues. In this regard, in recent years, there has been an increased demand for research and screening of new anti-cancer agents, including plant bioactive substances, that have few side effects during tumor management. About 35,000 phytoconstituents have been obtained from terrestrial and marine sources to be potentially employed in standard cancer therapies [2].

Several plant metabolites, such as flavonoids, phenols, terpenoids, saponins, and alkaloids, show chemoprotective activity toward cancer cells of different origins, resulting in cell cycle arrest or inducing apoptosis; they are also able to modulate microRNAs with oncosuppressive activity as well as inhibit oncogenes and anti-apoptotic factors [3]. The mechanisms of action of these bioactive compounds are various and may include the reduction of oxidative stress-mediated DNA damage and the promotion of apoptosis either by blocking checkpoints or by driving down the levels of anti-apoptotic proteins [4,5].

*Punica granatum* L. (pomegranate) belongs to the family Lythraceae (Punicoideae subfamily), and it is considered an ornamental tree indigenous to Mediterranean regions and Iran. Since ancient times, this plant has found application in traditional medicine, such as in the treatment of gastrointestinal, cardiovascular, and endocrine diseases. Indeed, the seed, flower, bark, peel, juice, and leaves of pomegranate are rich in potentially bioactive compounds, making it a functional food [6,7,8]. Numerous biological studies, focused on the phytocomplexes extracted from pomegranate fruit, have demonstrated the nutraceutical and nutritional properties of this plant extract. Acids, sugars, vitamins, minerals, and secondary metabolites, like simple phenols, flavonoids, anthocyanins, and tannins, are considered the main phytochemicals of pomegranate derivatives that present anti-inflammatory, antibacterial, anticarcinogenic, antihypertensive, and antiviral activities [9,10].

In particular, several studies have shown that the polyphenols contained in pomegranate exhibit anti-inflammatory effects and are nontoxic, demonstrating their possible adjuvant use in combination with standard anticancer therapies [11], just as the remarkable antioxidant capacity of the peel of the fruit of *P. granatum* has been shown to reduce the proliferation of breast cancer cells by inducing apoptosis [12].

In this study, phytocomplexes were isolated from pomegranate fruits using an extraction technique based on hydrodynamic cavitation (HC) processes, consisting in the creation of a periodic depression in a liquid mixture, for example by means of the circulation of the liquid through a nozzle of suitable geometric shape, such as an orifice plate or a Venturi tube. The sudden pressure decrease occurring in the accelerating flow forces the creation of vapor-filled nano- and micro-bubbles whenever the pressure falls below the vapor pressure. Then, the bubbles grow until the flow decelerates and enters an area of increased pressure, for example, in the divergence section of a Venturi tube, where the external force produces the sudden implosion of the bubbles. As a consequence of the bubble implosion, extremely intense pressure shock waves, hydraulic jets, transient heating, and chemical dissociation reactions occur [13]. In comparison with other conventional and emerging methods for the extraction of natural products, HC-based extraction showed generally higher effectiveness, efficiency, process yield, and straightforward scalability [14].

In a recent study, pomegranate extraction methods, either of whole fruit or residual fractions from juice squeezing, such as peel and seeds, were reviewed, finding that only a patented and protected method could afford industrial-scale production [15]. Conversely, either consolidated extraction methods, such as simple stirring, or emerging green methods, such as pressurized liquid extraction, ultrasound-assisted, and microwave-assisted extraction, also coupled with enzyme-assisted extraction, could not be proven at the industrial scale. However, ultrasound-assisted extraction, which was based on cavitation events in a liquid-solid mixture, showed good effectivity but only at the laboratory scale due to intrinsic scalability issues. It was concluded that research has not yet identified a sufficiently effective, efficient, affordable, and scalable technique for the extraction of whole pomegranate fruits, including the byproducts left after squeezing the juice, although the industry has progressed independently, offering important pomegranate-based ingredients widely used in food supplements.

Thus, based on the generally higher effectiveness, efficiency, and yield of HC-based extraction techniques observed with other natural products and fruits compared with other consolidated or emerging techniques, HC was suggested for the first time for the extraction of whole pomegranate fruits, following applications to citrus waste peel and other natural products [15].

Extracts collected at different times during the HC-based extraction process were characterized for their biochemical composition and antioxidant activity. The most antioxidant extract was selected for the in vitro assessment of its antiproliferative capacity against the human breast cancer line (AU565-PAR) as well as possible cytotoxic effects on peripheral blood mononuclear cells (PBMCs) from healthy donors.

## 2. Materials and Methods

### 2.1. Raw Materials and Extraction Process

Whole pomegranate fruits from the Wonderful variety were purchased at a local market in the Apulia region, Salento area (southeastern Italy in mid-November 2021, i.e., in the second half of the harvest period) and processed the day after.

In this study, the extraction of whole pomegranate fruits in water was performed using a semi-industrial-scale (200 L) HC pilot device optimized for food applications. The details of the HC-based extractor, comprising a closed hydraulic circuit with a centrifugal pump and a circular Venturi-shaped reactor as the key components, along with the meaning of the cavitation number as a measure of cavitation intensity and regime, were those described in a previous study [16]. No active heat dissipation method was applied. Power and energy consumption were measured by means of a three-phase digital power meter (IME, Milan, Italy, model D4-Pd). A total of 48 kg of whole fresh pomegranate fruit, including the thick peel, was ground in ice to coarse pieces with a maximum linear size of around 10 mm, then pitched in the HC device, where water was added until the volume including ice equaled 100 L. The initial temperature was 21.5 °C. After filtering with a 200 µm sieve (stainless steel mesh), five samples (M1–M5 fractions) were collected at different extraction times, temperatures, and specific energy consumption, as shown in Table 1, and stored in sterile bottles at −20 °C until analysis. The HC process was smooth, with a fairly constant cavitation number of 0.11 to 0.12, ensuring optimal cavitation yield [14].

All the original specimens were preserved at −80 °C at the Institute of Bioeconomy, National Research Council of Italy, Via Madonna del Piano 10, 50019 Florence, Italy, with the following vouchers: “ME1R” for raw material (whole pomegranate fruit), and “ME1-1”, “ME1-2”, “ME1-3”, “ME1-4,” and “ME1-5” for the filtered aqueous extracts (corresponding to fractions M1 to M5 in Table 1).

After thawing the extracts on ice, their volume was measured, and characteristics such as viscosity, color, and presence of particles/filaments were assessed. Based on the obtained volume, aliquots of samples were prepared and subjected to the following analyses: in detail, pure extracts were kept as they were (crude; C), freeze-dried and resuspended in water (lyophilized; L), and filtered with nitex (0.45 μm) to produce the aqueous extract (aqueous; A).

### 2.2. Total Phenolic Content Analysis

A Folin-Ciocalteu spectrophotometric assay was performed to determine the amount of simple phenolic compounds present in the extracts, following the method of Impei and colleagues [17]. After incubation with the reagents, an ELISA microplate reader (Sunrise, Tecan, Austria) was used to read the absorbance of the samples at 760 nm. The concentration of the phenolic component was estimated by a calibration curve (0–3 mg/L; R^2^ = 0.9975), adequately prepared using gallic acid (GA) as a standard. Results were expressed as micrograms (µg) of gallic acid equivalent per milligram (mg) of sample fresh weight (µg GAE/mg FW).

### 2.3. Total Flavonoid Content Analysis

The aluminum chloride colorimetric method was applied to quantify flavonoids, as described by Di Marco and colleagues [18]. For this assay, the absorbance of the samples was measured at 415 nm (using the same ELISA microplate reader), and the concentration of the total flavonoid content was calculated based on a standard curve of quercetin (Q; 0–5 mg/L; R^2^ = 0.9975). Results were reported as µg of quercetin equivalent per mg of sample fresh weight (µg QE/mg FW).

### 2.4. Determination of Tannin Content

The concentration of tannins in the samples was determined spectrophotometrically, using the method described by Weidner et al. [19] and following the modifications reported in Impei et al. [17]. The absorbance of the sample was measured at 510 nm. Pure catechin was used for the preparation of the standard calibration curve (C; 0–300 mg/L; R^2^ = 0.9891). The amount of total tannins was expressed as µg of catechin equivalent per mg of sample fresh weight (µg CE/mg FW).

### 2.5. Determination of Total and Specific Anthocyanins by Spectrophotometric Analysis

According to the Giusti and Wrolstad method [20], two buffers (0.025 M potassium chloride buffer, pH 1.0; 0.4 M sodium acetate buffer, pH 4.5) were prepared. Fifty µL of each of them were separately mixed with 50 µL of the extract. Then, the samples were incubated in the dark for 15 min. The absorbance of both solutions was measured at 530 nm (for total anthocyanins), 510 nm (for cyanidin), 523 nm (for delphinidine), 557 nm (for malvidine), 505 nm (for pelargonidin), 496 nm (for pelargonidin 3-glucoside), 511 nm (for peonidin), 520 nm (for petunidin 3-glucoside), and 700 nm (as background) by the Varian Cary 50 Bio UV-Vis spectrophotometer (GEMINI Lab sustainable equipment, Apeldoorn, The Netherlands). Equation (1) was used for measuring the absorbance of the sample:A = [(Ax − A700) pH 1.0 − (Ax − A700) pH 4.5],(1)
where x is the wavelength at which each single anthocyanin absorbs. Subsequently, the concentration of the single anthocyanin was estimated according to Equation (2):C = (A × MW × dilution factor × 1000)/(ε × l),(2)
where A is the absorbance measured for the sample at the wavelength of the single anthocyanin, MW is the molecular weight of the single anthocyanin (i.e., cyanidin-3-glucoside as standard equivalent for the quantitation of the total anthocyanin, 449.2; cyanidin, 287.24; delphinidine, 303.24; malvidine, 331.29; pelargonidin, 271.24; pelargonidin 3-glucoside, 433.4; peonidin, 301.27; petunidin 3-glucoside, 479.41), ε is the molar absorptivity of the anthocyanin (i.e., cyanidin-3-glucoside, 26,900; cyanidin, 24,600; delphinidine, 34,700; malvidine, 36,200; pelargonidin, 17,800; pelargonidin 3-glucoside, 15,600; peonidin, 37,200; petunidin 3-glucoside, 18,900), and l is the length of the cuvette employed for the realization of the test. Results were reported as µg of cyanidin-3-glucoside equivalent per mg of sample fresh weight (µg Cy3GE/mg FW) for total anthocyanins and µg of each anthocyanin per mg of sample fresh weight (µg/mg FW) for single anthocyanins.

### 2.6. In Vitro Antiradical Assay

FRAP (2,4,6-tris 2-pyridyl-s-triazine; Sigma Aldrich) and ABTS (2,2′-azino-bis-3-ethylbenzothiazoline-6-sulfonic acid; Sigma-Aldrich) assays were carried out according to Benzie and Strain [21] and Re et al. [22] guidelines after the application of the modifications reported in Gismondi et al. [23]. These tests, based on different biochemical principles, were carried out to determine the antiradical potential of each extract. In detail, the FRAP assay is a colorimetric method based on the reduction of the Fe^3+^ TPTZ colorless complex to the Fe^2+^ TPTZ colored complex in the presence of antioxidant molecules. On the other hand, the ABTS spectrophotometric assay measures the ability of the antioxidants present in an extract to scavenge the green solution containing the stable radical cation ABTS+. Thus, the phytochemicals would be able to inhibit the formation of the colored ABTS radical. Free radical scavenging activity was expressed as µg of ascorbic acid equivalents per mg of sample fresh weight (µg AAE/mg) for the FRAP test and as micromoles of ascorbic acid equivalents per mg of sample fresh weight (µmol AAE/mg) for ABTS, according to adequate calibration curves produced with pure ascorbic acid.

### 2.7. HPLC-DAD Analysis

To delineate the phytocomplex of each extract, a high pressure liquid chromatographic system (HPLC) was employed for the analysis. The system was provided with a CBM-20A controller, an LC-20 AD pump, a SIL-20a HT autosampler, and an SPD-M20A diode array detector (DAD) (Shimadzu, Kyoto, Japan). A Luna 3u C18(2) column (150 mm × 4.60 mm × 3 μm) (Phenomenex, Torrance, CA, USA) and two mobile phases consisting of 1% formic acid (*v*/*v*) (phase A) and MeOH (phase B), at a flow rate of 0.95 mL per minute, were applied for the chromatographic separation. Each sample (20 μL) was injected into the instrument, and the column temperature was set at 40 °C. The elution began at 15% B solvent and was maintained for 20 min; after that, the solvent B was linearly increased up to 35% in 20 min and up to 90% in 55 min; at 70 min, the pump B value was reported at the initial condition. Data acquisition was carried out using LAB-SOLUTION software (Shimadzu). Flavonoid compounds (i.e., resveratrol; quercetine-3-glucoside; myricetin; quercetin; genistein; kaempferol; chrysin; epicatechin) and phenolics (i.e., gallic acid; 3-hydroxytyrosol; vanillic acid; rosmarinic acid; 4-hydroxybenzoic acid; chlorogenic acid; caffeic acid; syringic acid; ρ-coumaric acid; salicylic acid; 1,1-dimethylallyl caffeate; caffeic acid phenethyl ester; 5,7-dimethoxycumarin) were quantified in the samples at 280 nm. Each metabolite was analyzed in qualitative and quantitative terms, comparing their retention times (minutes), absorption spectra, and peak areas with those of pure standard molecules (Sigma-Aldrich, Milan, Italy). Results were reported as ng of standard equivalent per mg of fresh sample weight.

### 2.8. Cell Culture

In this study, the human breast cancer cell line AU565 was used (kindly provided by Professor Ira-Ida Skvortsova, Medical University of Innsbruck Tyrolean Cancer Research Institute, Innsbruck, Austria) and maintained at 37 °C in a humidified 5% CO_2_ atmosphere in RPMI-1640 medium supplemented with 10% fetal bovine serum (FBS), L-glutamine (2 mM), penicillin-streptomycin (100 mg/mL) (all from Sigma, St. Louis, MO, USA). In addition, cytotoxic effects on peripheral blood mononuclear cells (PBMCs) from healthy donors (HD) were evaluated to identify a range of non-toxic concentrations for healthy cells that could have antiproliferative and proapoptotic effects on tumor cells, as demonstrated in previous works on other natural substances [24,25,26,27].

Three HD were obtained from individuals attending the local blood Transfusion Unit of Policlinic of “Tor Vergata” in Rome who were referred to the Virology Unit of PTV for screening and provided written informed consent. The study was performed in accordance with the ethical principles of the Declaration of Helsinki and the Guidelines for Good Clinical Practice. The ethical committee of Tor Vergata University/Hospital approved this study (protocol number: COVID_SEET Prot. No. 7562/2020).

PBMCs from heparinized blood samples were isolated by density gradient centrifugation (Pancoll human, PAN-Biotech, Aidenbach, Bavaria, Germany) and cultured at a density of 0.1 × 10^6^/500 µL in RPMI 1640 (PAN Biotech, Aidenbach, Bavaria, Germany) enriched with 2 mM L-glutamine, 100 U/mL penicillin, 0.1 mg/mL streptomycin, and 12% fetal bovine serum in the presence of recombinant human interleukin-2 (IL-2) 20 U/mL (all from Sigma, St. Louis, MO, USA).

### 2.9. M3 Treatments

To verify the effect on PBMCs and AU565 of M3 extract from the L series, which was chosen due to its high content in secondary metabolites among all samples, cells were cultured at a concentration of 0.03 × 10^6^/500 µL for 18 h at 37 °C in a humidified atmosphere containing 5% CO_2_. Then, they were treated, or not, with the M3-L sample at different concentrations, starting from 0.06 mg/mL to 50 mg/mL, for 48 h. In addition to the control conditions, a treatment was also carried out with ddH_2_O, as a control vehicle for M3, and with etoposide, an inhibitor of DNA synthesis, forming a complex with topoisomerase II as a positive control of cell death [28,29,30]. After incubation, cells were detached with a trypsin solution and analyzed. Similarly, PBMCs were treated, or not, with M3 at the same concentrations reported above for the same timing.

### 2.10. Cell Viability and Apoptosis Assay

Alive and dead cells were counted by optic microscopy at 48 h of treatment with Trypan Blue (EuroClone S.p.A., Pero, Italy). The percentage of apoptotic cells was analyzed by flow cytometry (CytoFLEX; Beckman Coulter, Inc., Brea, CA, USA), which measures the number of hypodiploid nuclei in 150.000 events. In addition, early apoptosis by annexin V (ANX-V)-7-aminoactinomycin D (7AAD) vitality marker staining (EuroClone S.p.A., Pero, Italy) was evaluated after 4 h of treatment. Data acquisition and analysis were performed by CytExpert 2.4 (Beckman Coulter, Inc.).

### 2.11. Reactive Oxygen Species (ROS) Production Assay

ROS production was evaluated by the DCFDA/H2DCFDA-Cellular ROS Assay Kit (ab113851, Abcam, Cambridge, UK). The DCFDA assay protocol is based on the fluorescence of this marker after interaction with ROS, considering an excitation/emission spectrum fixed at 485 nm/535 nm. After 48 h of treatment, detached AU565 cells were incubated for 30 min at 37 °C with 20 µM DCFDA and then washed with 1× Buffer according to the manufacturer’s instructions. The analysis was carried out by flow cytometry (CytoFLEX; Beckman Coulter, Inc.). Alive cells were selected using a morphological gate, and the mean fluorescence intensity (MFI) of DCFDA was assessed in relation to the different treatments, as shown in Appendix A.

### 2.12. Statistical Analysis

The results were reported as means ± standard deviation (SD) of 3 independent measurements and technical duplicates (n = 6) or 2 independent in duplicates (n = 4). Data were subjected to a one-way analysis of variance (ANOVA) and a post-hoc lowest standard deviations (LSD) test (Excel software); *p* values were indicated as follows: * *p* < 0.05; ** *p* < 0.01; *** *p* < 0.001. The relationship existing between classes of phytochemicals and antioxidant power was estimated by Pearson’s correlation coefficients, using PAST software (v4.03).

## 3. Results

### 3.1. Biochemical Characterization of Pomegranate Extracts

The contents of total phenols, flavonoids, tannins, and anthocyanins of M1–M5 fractions from crude (C), lyophilized (L), and aqueous (A) pomegranate extracts were analyzed by means of spectrophotometric analyses (Figure 1, Appendix A). Considering all the studied samples, the extreme values for the total phenols were 0.079 µg GAE/mg in M1 from A and 0.387 µg GAE/mg in M3 from L (Figure 1, panel a), while those for flavonoids were 1.756 µg QE/mg in M1 from A and 2.889 µg QE/mg in M5 from L (Figure 1, panel b). Tannins ranged from 0.366 µg CE/mg in M5 from L to 0.722 µg CE/mg in M5 from A (Figure 1, panel c), while anthocyanins ranged from 2.20 µg Cy3GE/mg in M1 from A to 13.92 µg Cy3GE/mg in M4 from L (Figure 1, panel d). In general, for total phenols, we observed lower values in all fractions of A compared to C and L, albeit not significantly. C and L, instead, always showed more similar values in all M fractions. For flavonoids, we did not observe a specific trend, but the samples from A presented the lowest values, like total phenols; on the other hand, for the tannins, we could see that all fractions from A had significantly higher values compared to C and L. Tannins are polyphenolic compounds usually associated with seeds, woody tissues, leaves, roots, and fruit peel, which participate in determining the bioactivity, strange smell, and astringent taste of plant extracts. They can be classified as hydrolysable, condensed, and complex tannins on the basis of their solubility and chemical structure, and, in plants, they play a fundamental role as deterrents to herbivory and antimicrobial agents. Finally, our data showed a high level of total anthocyanins in all C and L samples, particularly in the fractions M3 and M4.

Via spectrophotometric assays, the content of 7 anthocyanins in the various fractions (M1–M5) from C, L, and A extracts was also evaluated (Figure 2, Appendix A). Anthocyanins are plant secondary metabolites belonging to the flavonoid class. In pomegranate, as in other plants, they are responsible for the coloration of fruits, flowers, and leaves and, consequently, useful in pollination and seed dispersal, but they are also involved in the prevention of photo-oxidative damage and in the response to various abiotic stresses. Additionally, anthocyanins, such as cyanidin, delphinidin, and pelargonidin, are among the bioactive phytochemicals that may have beneficial effects on human health. In particular, cyanidin is the most abundant anthocyanin in fruits and vegetables, and it is considered a potent chemopreventive agent. From our data, cyanidin showed similar values in the three extracts at all extraction times, while the amount of delphinidin, peonidin, pelargonidin, pelagordinin-3-glucoside, and petunidin-3-glucoside tended to be overlapping and comparable for L and C in all fractions but lower than the respective A fractions. An exception was represented by pelagordinin-3-glucoside, whose measurements appeared more variable in C and L procedures at different time points. It was also possible to observe an opposite trend for malvidin, whose C values were higher than those of the respective L and A fractions.

HPLC-DAD was used to evaluate and quantify 21 specific plant secondary metabolites, such as phenolics and flavonoids (Appendix A). Gallic acid, genistein, 1,1-dimethylallyl caffeate, and chrysin were not detected in any pomegranate sample. Aqueous fractions, if compared at all time points from C and L extracts, lacked four molecules: 3-hydroxytyrosol, quercetin, 5,7-dimethoxycoumarin, and kaempferol. Other compounds (e.g., caffeic acid phenethyl ester, kaempferol, quercetin, and 5,7-dimethoxycoumarin) were detected in traces in the C and L fractions (that is always <1.5 ng/mg of plant fresh weight) and only at late extraction times (i.e., M3–M5). Rosmarinic acid and myricetin showed low values, compared to the other analytes, in all samples (with a maximum of 6.82 ng/mg in M2 from L); in addition, these molecules were not detectable in M1 from A samples. Generally, the highest values of phytochemicals, among all samples, were achieved by epicatechin (maximum values of 606.14 ng/mg in M3 from C, 535.67 ng/mg in M5 from L, and 642.98 ng/mg in M3 from A), 3-hydroxytyrosol (maximum values of 556.45 ng/mg in M4 from C, and 475.93 ng/mg in M3 from L), chlorogenic acid (maximum values of 263.73 ng/mg in M2 from C, 228.93 ng/mg in M3 from L, and 358.02 ng/mg in M4 from A), quercetin-3-O-glucoside (maximum values of 223.11 ng/mg in M3 from C, 189.47 ng/mg in M3 from L, and 275.08 ng/mg in M4 from A), 4-hydroxybenzoic acid (maximum values of 129.56 ng/mg in M4 from C, 160.67 ng/mg in M5 from L, and 217.31 ng/mg in M1 from A), and salicylic acid (maximum values of 188.72 ng/mg in M3 from C, 135.93 ng/mg in M5 from L, and 172.69 ng/mg in M3 from A). Salicylic acid tended to be less concentrated in all lyophilized fractions with respect to the C and A ones, ranging from 4.93 ng/mg in M5 from L to 188.72 ng/mg in M2 from C. Resveratrol reached its highest concentration in M3 in L. Among the phenolics, two molecules were found, albeit in traces, only in some fractions: 5,7-dimethoxycoumarin in M4 and M5 from C (at 0.26 ng/mg and 0.47 ng/mg, respectively) and M4 and M5 from L (at 0.29 ng/mg and 0.42 ng/mg, respectively), and caffeic acid phenethyl ester in M3, M4, and M5 from C (at 0.13 ng/mg, 0.29 ng/mg, and 0.27 ng/mg, respectively), M5 from L (at 0.15 ng/mg), and M1 from A (at 1.46 ng/mg). In the aqueous fractions, p-coumaric acid was detected only in M4 and M5. Overall, the samples presenting the highest values of the investigated molecules were the M3 fractions from C, L, and A extracts, followed by M5 from L and M4 from A.

The antioxidant activity of pomegranate samples was measured by ABTS and FRAP assays. The results obtained for each experimental point were indicated as spots in the chart area of panel A in Figure 3. Here, each spot was obtained by correlating the respective ABTS and FRAP results. The C and L samples showed high antiradical power during both tests. The mean value of the FRAP assay for all fractions of these two extracts was overlapping (i.e., 274.8 µg AAE/mg for C and 275.2 µg AAE/mg for L); similarly, this occurred also for ABTS (462.77 µmol AAE/mg for C and 462.65 µmol AAE/mg for L). Indeed, the samples tended to be grouped in the upper right quadrant of the graph shown in Figure 3. Differently, the fractions of the aqueous extract had high ABTS values but low FRAP values; anyway, they could be clearly distinguished in the graph by their respective ABTS values. Among all, M4 from A extract showed the best antiradical power (i.e., 94.55 µg AAE/mg for FRAP and 432.92 µmol AAE/mg for ABTS). In order to understand which group of secondary metabolites from pomegranate mainly influenced the antioxidant potential of the extracts, Pearson’s correlation analysis was carried out. The results indicated a positive correlation between antioxidant effects (measured by ABTS or FRAP) and phenolics (i.e., simple phenols and flavonoids). Among the latter, the class of anthocyanins seemed to be particularly responsible for the antiradical power of the extracts. By contrast, a negative link existed between ABTS/FRAP and tannins (Figure 3, panel b).

### 3.2. Effects of M3 Treatment on Growth Inhibition in Cancer Cells and PBMCs

Based on the higher concentration of phytochemicals in M3 from the L series, the biological effect of this pomegranate sample on AU565 and PBMC growth and viability was assessed. After 48 h of exposure, increasing doses of M3 (starting from 3.75 mg/mL) significantly inhibited AU565 growth compared to CTR, and the same result was found with the etoposide treatment (ETO). Compared to AU565, PBMCs appear to be susceptible to M3, but a drastic reduction in cell growth was neither evident nor statistically significant, as well as comparable with etoposide treatment (Figure 4). The cytotoxicity of the treatments was also evaluated in terms of effective concentration (30%) (EC30) and lethal dose (30%) (LD30) (Table 2).

### 3.3. Effects of M3 Treatment on the Induction of Apoptosis in Cancer Cells and PBMCs

To test whether M3 treatments could affect cell death in the investigated cancer cell line, an analysis of apoptosis was performed by assessing the content of hypodiploid DNA using Propidium Iodide staining and analysis by flow cytometry. Histograms in Figure 5 represent the fold change of dead cells per each concentration of treatment compared to their control. Increasing doses of M3 induce apoptosis in AU565, in particular M3 in a range of concentration from 5 to 25 mg/mL (*p* < 0.050) and 50 mg/mL (*p* < 0.01) induced apoptosis in a statistically significant way compared to the control. Interestingly, there was no difference in apoptosis rate for PBMCs. An inverse and significant correlation was observed between the percentage of apoptosis and the viability analyzed with trypan blue exclusion test (Spearman’s *ρ* = −826, *p* < 0.001).

To confirm this observation, the effect of M3 on cell death was investigated via staining of intact cells with the ANX-V/7AAD after 4 h after treatment. This method enables the distinction of early apoptotic cells (ANX-V+/7AAD−) from late apoptotic/necrotic ones (ANX-V+/7AAD+). M3 at 25 mg/mL induce early apoptosis and late apoptosis/necrosis compared to their control, as shown in Figure 6 and Table 3.

### 3.4. Effects of M3 Treatment on the Modulation of ROS Production in Cancer Cells

Analysis of ROS production assessed by ab113851 DCFDA/H2DCFDA in AU565 cells showed a high median fluorescence intensity in the control conditions, as represented in Figure 7. Treatment with increasing doses of M3 induced a significant (*p* < 0.01) reduction in median fluorescence intensity at 3.75 mg/mL and highly significant (*p* < 0.001) at 5 mg/mL and 10 mg/mL.

## 4. Discussion

In *P. granatum*, the availability of appreciable levels of nutraceuticals, such as polyphenols, tannins, and anthocyanins, fostered interest in the consumption of this functional food with significant health-promoting properties [31]. In our study, the total content of phenols, flavonoids, and anthocyanins, measured by spectrophotometric assays, was constant across C and L extracts, while the same classes of secondary metabolites tended to decrease in the A fractions, albeit not significantly, perhaps due to the filtering procedure applied to these samples. In general, the fractions that retained greater amounts of phytochemicals were those from the C and L series and, in particular, M2, M3, and M4 for total phenols, M3, M4, and M5 for flavonoids, and M3 and M4 for anthocyanins. By contrast, all aqueous extracts were significantly rich in tannins. However, according to the results of Pearson’s correlation, this class of metabolites negatively influenced the antiradical power of the extracts. Tannins form complexes with proteins, enzymes, and polymers, such as cellulose and hemicelluloses. The stability of the tannin-protein complex depends on several factors that are dynamic and time-dependent [32]. It is possible that the filtering performed on the A samples eliminated the protein component of the pomegranate, allowing the release of the tannins present in the extracts. A portion of these tannins could be correlated with epicatechin, whose detection by HPLC-DAD revealed the highest values precisely in the aqueous fractions. Indeed, catechins are considered precursors of tannins [32].

The level of total anthocyanins was measured as they represent typical components of the pomegranate, justifying part of the antioxidant activity of its extracts [33]. Indeed, the content of these secondary metabolites was higher than that of the other molecular classes. In parallel, we tried to quantify the concentration of single anthocyanins in the samples, selecting those more abundant in *P. granatum*. Overall, the samples richest in anthocyanins were M3 and M4 for all the extracts, as well as for total phenol, flavonoid, and tannin content.

In order to obtain a fingerprint of the extracts, specific phenolics and flavonoids were monitored by HPLC-DAD. Interesting was the absence of gallic acid in all the samples, although it represents a peculiar unit of the tannins [34]. Thus, this simple phenol could have been degraded during the cavitation process or was absent in its free form. In the aqueous extracts, vanillic acid, epicatechin, and caffeic acid showed the highest values of the dataset in M3, as well as chlorogenic acid and quercetine-3-*O*-glucoside in the M4 fraction. Overall, the chromatographic analysis induced us to reflect on the fact that A samples lose some phenolics (e.g., quercetin, 5,7-dimethoxycoumarin), suggesting that such a type of preparation was not optimal for preserving antioxidant phytochemicals. This observation was also confirmed by the phenols, flavonoid, and especially anthocyanin content, which characterized the biochemical profiles of the C and L fractions and reflected the high antioxidant activity of these samples, as detected by means of FRAP and ABTS tests and validated by Pearson’s correlation analysis. It is noteworthy that just 5 min of process time (sample M1) were enough to produce relatively high contents of bioactive compounds, likely due to the immediate extraction of pomegranate juice. The apparent, slight decline in the content of bioactive compounds for samples collected after M3 or M4 could suggest a degradation of certain heat-sensitive compounds, unbalanced by further extraction from peel and seeds. Finally, M3 was chosen instead of M4 because the respective contents of bioactive compounds, as well as the levels of antioxidant activity, were indistinguishable. M3 was produced with only 15 min of process time (against 25 min for M4) at a temperature of about 30 °C (36 °C for M4) and much lower energy consumption (38 Wh per kg of fresh raw material) than M4 (62 Wh/kg), thus allowing higher affordability and productivity for the HC-based extraction method.

Several research works have provided ample evidence on the antiproliferative and proapoptotic properties of pomegranate products, such as the whole fruit, its juice, and oil, which exert anti-inflammatory, antiproliferative, and anti-tumor effects in different types of tumors via the modulation of multiple signaling pathways, such as NFkB, COX2 phosphorylation expression of STAT3, and AKT [35,36,37,38,39].

In this study, preliminary results on the proapoptotic capacity of an HC-based extract of whole pomegranate fruit were evaluated. As described in other studies [11,12] pomegranate contains a high amount of polyphenols with anti-inflammatory and therapeutic effects on different types of breast cancer, suggesting its use in combination with standard therapies. We observed that the M3-L sample reduced cell proliferation and induced apoptosis in the AU565 breast cancer line. In particular, the in vitro experiments showed that starting at 3.75 mg/mL of M3, a significant decrease in the viability of tumor cells was detectable, along with an enhancement of cell death. Interestingly, the same cellular phenomena in PBMCs appeared to be comparable to those of the untreated control at all concentrations tested, confirming that only cancer cells were susceptible to pomegranate exposure.

To confirm that the treatments induced apoptosis, an annexin V/PI assay was carried out on AU565 and PBMCs, distinguishing early apoptotic cells from late apoptotic/necrotic ones [25]. The treatment with M3 determined a significant increase in the percentage of annexin V-positive cells compared to the control at 25 mg/mL. Moreover, at this concentration, the levels of late apoptotic/necrotic cells were higher compared with the untreated sample. Clearly, these experiments asserted that plant treatment was able to specifically induce cell death by apoptosis. Finally, the intracellular ROS concentration was also measured, showing a significant decrease in the oxidative status after treatment with the pomegranate sample.

Although any inference on the exact mechanisms leading to the observed antiproliferative effects would be premature, it is worth noting that the pomegranate fruit stands out for its antioxidant capacity. For example, pomegranate juice has the highest antioxidant capacity compared to other polyphenol-rich beverages, such as red wine, grape juice, blueberry juice, and green tea [40]. It is indeed known that the considered pomegranate extract contains a broad spectrum of nutraceuticals with strong antioxidant capacity. These compounds would exert their antiradical activity in several ways, including by removing or neutralizing free radicals, chelating metals, influencing cell signaling pathways, and modulating gene expression [41,42]. Studies on Caco-2 colon cancer cells treated with punicalagin and ellagic acid, obtained from pomegranate, have shown pro-apoptotic and oxidative stress-reducing effects, providing information about the possible molecular mechanism of action underlying the bioactivity of *P. granatum* [43].

## 5. Conclusions

According to all the previous data and taking into consideration the potential use of plant derivatives as food supplements with healthy effects, the lyophilized form of the extract from whole pomegranate fruit obtained using HC would represent the best solution to preserve intact antioxidants and related antiradical properties, considering its stability due to the lack of water responsible for oxidative reactions degrading phytochemicals. Likely acting via the reduction in oxidative stress, which in turn contributes to cancer development, pomegranate-derived antioxidant phytochemicals obtained using HC during just 15 min of process time and with very low energy consumption (38 Wh per kg of fresh raw material) showed significant anticancer activity and safety towards healthy cells.

While it has long been known that pomegranate extracts possess anticancer activity, the main point of this study was the proof that extracts obtained using hydrodynamic cavitation, a feasible, effective, efficient, green, and scalable extraction technique of natural products, were endowed with potentially remarkable anticancer activity, paving the way to an enabling technology for a more affordable exploitation of the whole pomegranate fruit.

## Figures and Tables

**Figure 1 antioxidants-12-01560-f001:**
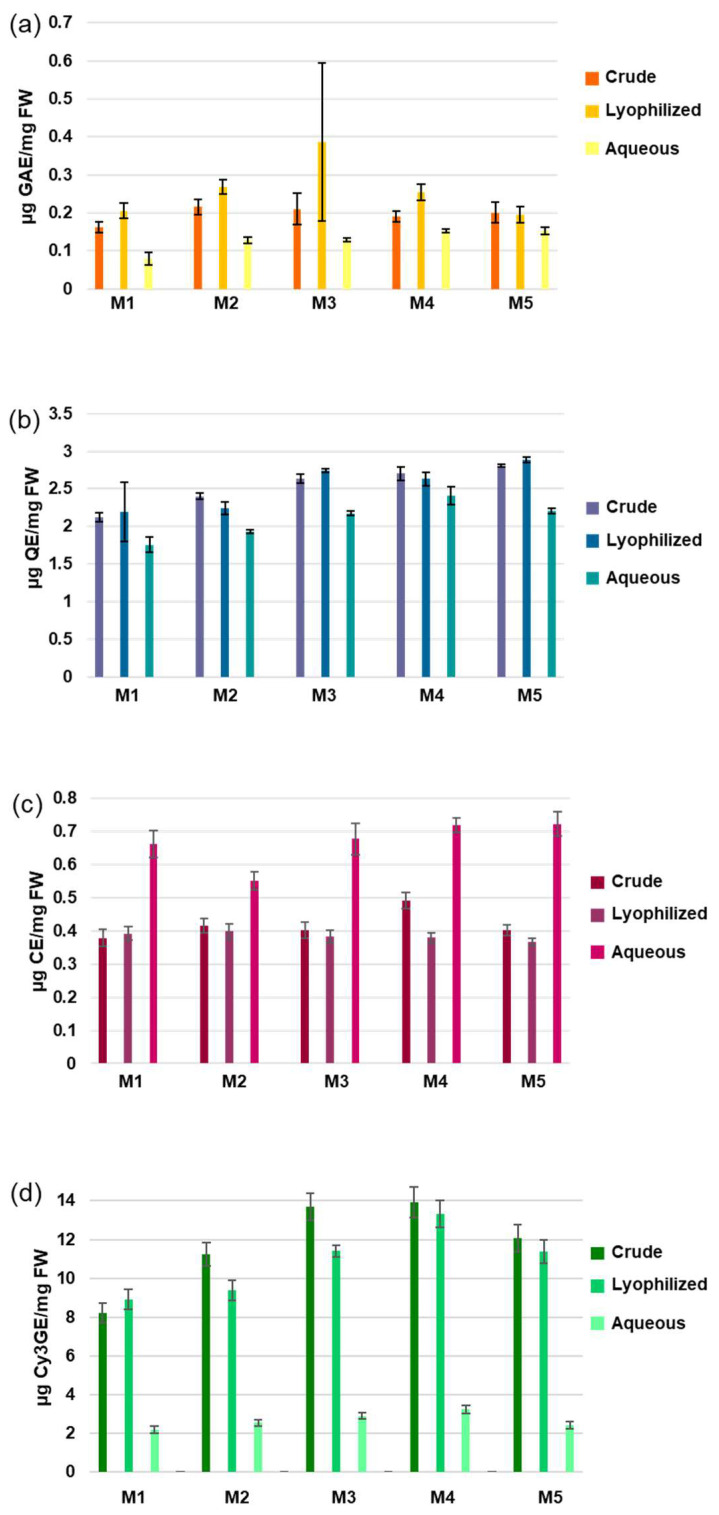
Content of secondary metabolites in pomegranate samples. (**a**) Total content of phenolics; (**b**) Total content of flavonoids; (**c**) Total content of tannins; (**d**) Total content of anthocyanins. These quantities were measured by spectrophotometric assays in the fractions (M1–M5) of crude, lyophilized, and aqueous extracts. Results are reported as mean ± S.D. (n = 3). The unit of measure was indicated in the *y*-axis of each graph. Only in (**c**) and in (**d**) the significance of the results showed a *p*-value < 0.05 for all the aqueous samples vs. the respective crude and lyophilized ones.

**Figure 2 antioxidants-12-01560-f002:**
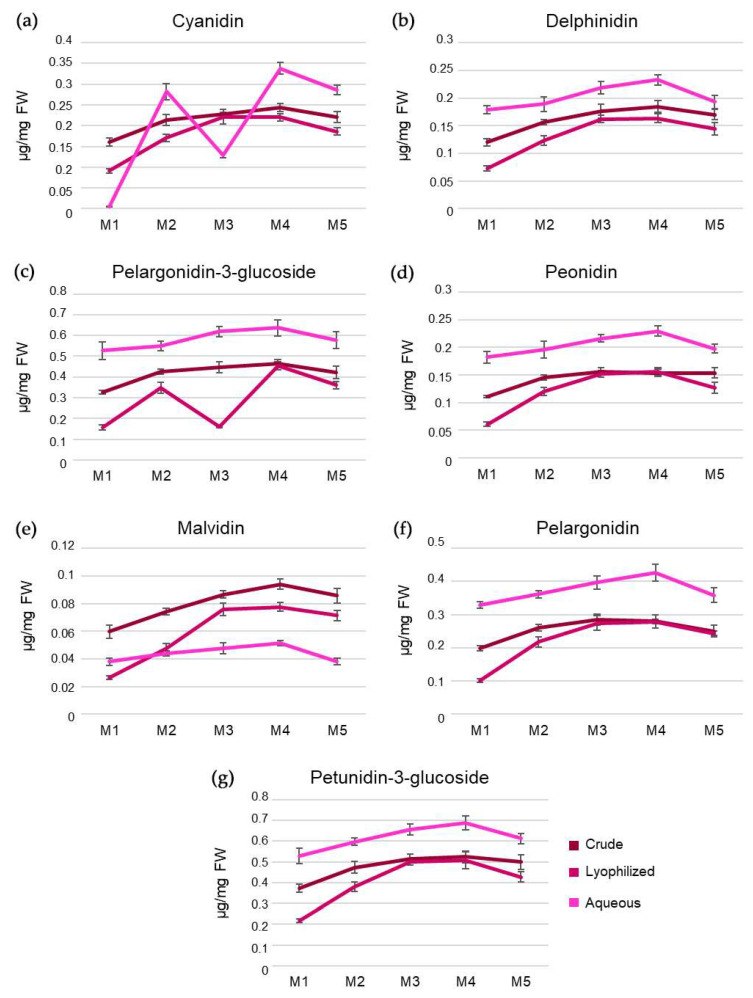
Anthocyanin quantitation in pomegranate samples in the fractions (M1–M5) of crude, lyophilized, and aqueous extracts. Line charts show the level of (**a**) cyanidin; (**b**) delphinidin; (**c**) pelargonidin 3-glucoside; (**d**) peonidin; (**e**) malvidin; (**f**) pelargonidin; (**g**) petunidin 3-glucoside. Results were expressed as µg per mg of fresh plant weight and reported as mean ± S.D. (n = 3).

**Figure 3 antioxidants-12-01560-f003:**
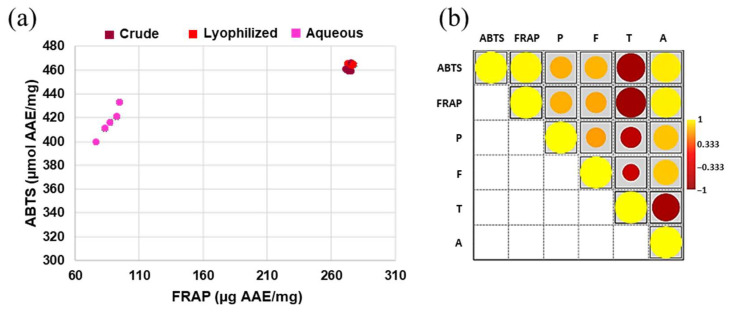
Antioxidant activity of pomegranate samples. (**a**) Antiradical power measured by ABTS and FRAP assays in the fractions (M1–M5) of crude, lyophilized, and aqueous extracts. In the graph, in order to correlate the two tests, the units of measure for FRAP results were indicated on the *x*-axis (i.e., the antioxidant power was expressed as µg AAE per mg of fresh plant weight). On the *y*-axis, the units of measure for ABTS data were reported (as µmol AAE per mg of fresh plant weight). (**b**) Pearson’s correlation between content of plant secondary metabolites and antioxidant potentials (*p* < 0.05 boxed). Legend: phenols (P), flavonoids (F), tannins (T), anthocyanins (A).

**Figure 4 antioxidants-12-01560-f004:**
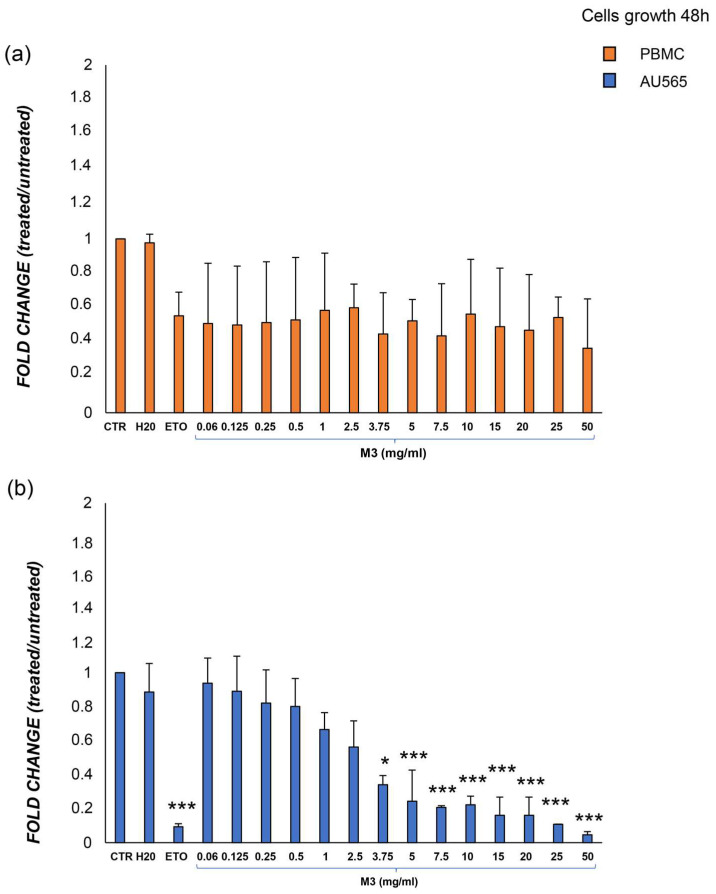
Viability analysis following treatment with increasing doses of M3. (**a**) PBMC from healthy donors; (**b**) AU565 cells. Results are expressed as the ratio between the number of alive cells after treatments and that measured in the control condition (fold change). (*) *p* ≤ 0.050; (***) *p* < 0.001. Data are shown as mean ± SD of at least three independent experiments performed in technical duplicates (n = 6).

**Figure 5 antioxidants-12-01560-f005:**
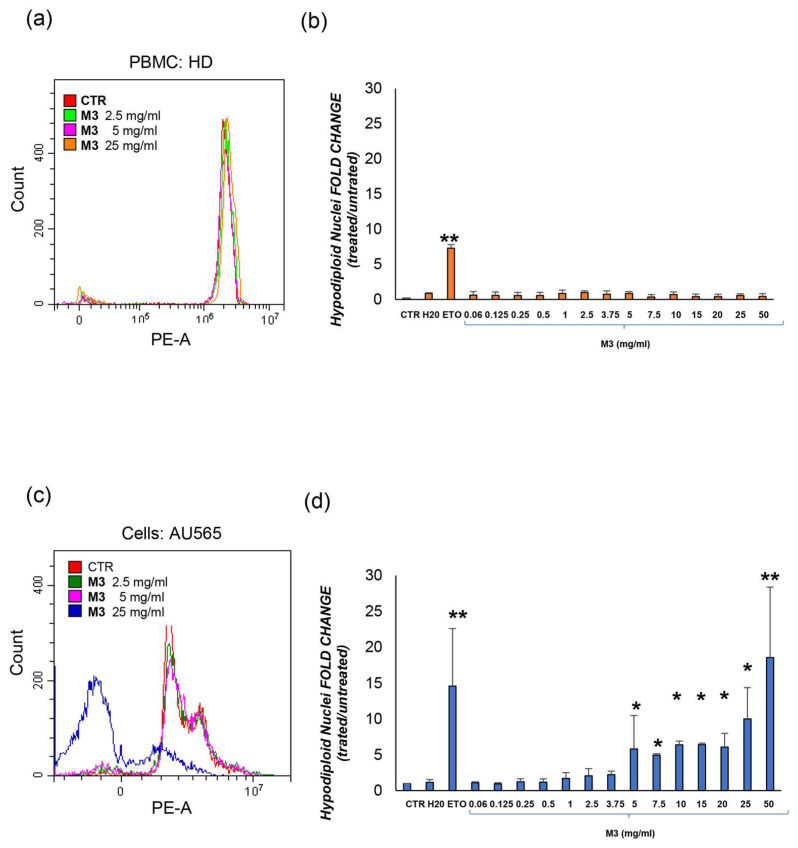
Evaluation of the apoptosis expressed as mean fluorescence intensity in (**a**,**b**) PBMC; (**c**,**d**) AU565 cells. The results in the histograms are expressed as the ratio between the percentage of hypodiploid nuclei measured in the treatment and control conditions (fold change). (*) *p* ≤ 0.050; (**) *p* ≤ 0.010. Data are shown as mean ± SD of at least three independent experiments performed in technical duplicates (n = 6).

**Figure 6 antioxidants-12-01560-f006:**
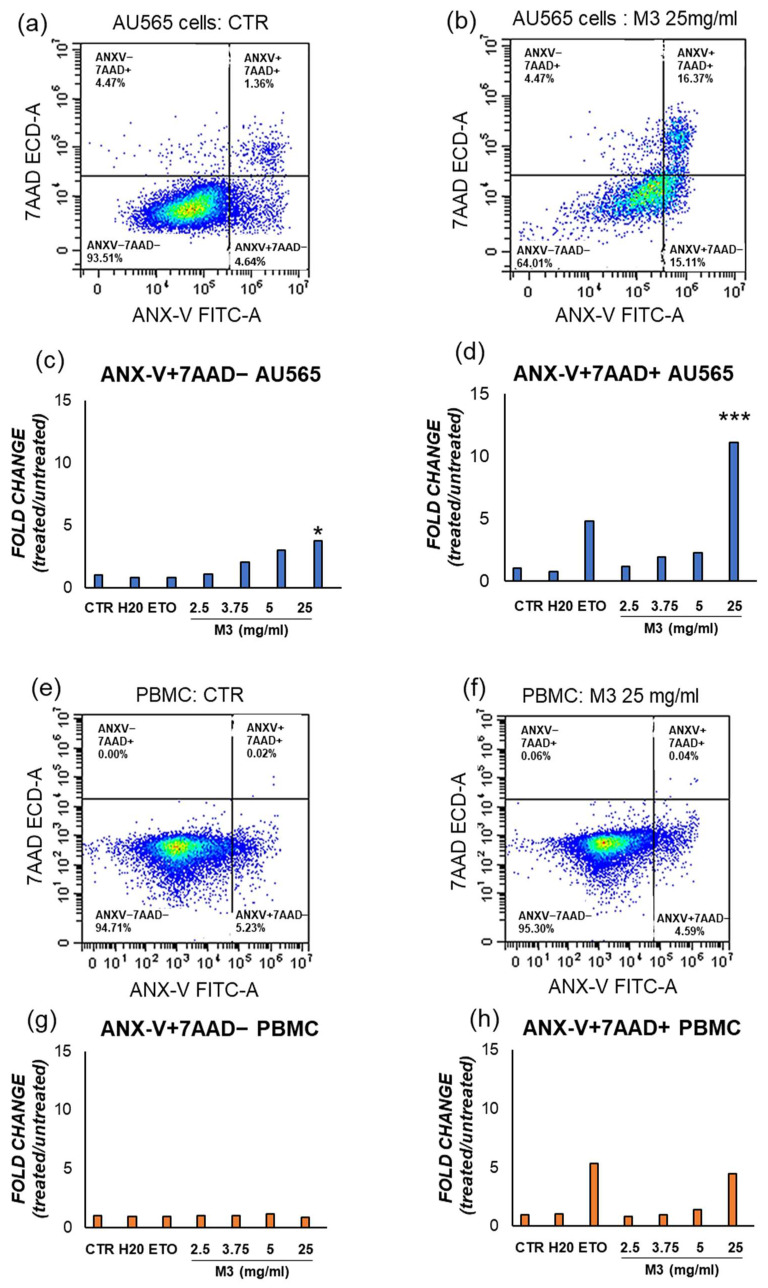
Representative flow cytometric analysis of the effect of M3 25 mg/mL on early apoptosis (ANX-V+/7AAD−) and late apoptosis/necrosis (ANX-V+/7AAD+) in (**b**) AU565; (**f**) PBMC, with respect to their control conditions (**a**,**e**). The results in the histograms represent the fold change of the ratio between the percentage of cells ANX-V+/7AAD− and ANX-V+/7AAD+, respectively in (**c**,**d**) AU565; (**g**,**h**) PMBCs. (*) *p* ≤ 0.050; (***) *p* < 0.001. Data are shown as mean ± SD of at least two independent experiments performed in technical duplicates (n = 4).

**Figure 7 antioxidants-12-01560-f007:**
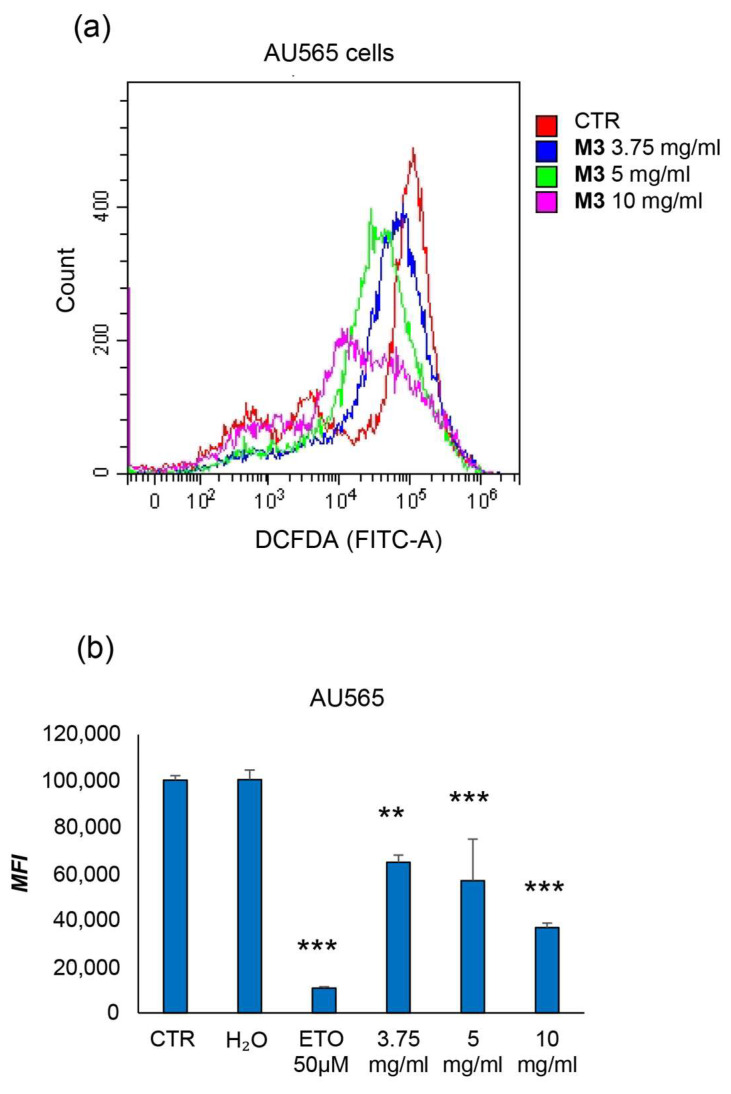
Evaluation of effects of M3 treatment on the modulation of ROS production in AU565. (**a**) Results are expressed as cell counts in relation to the fluorescence intensity of the DCFDA probe (FITC-A) following treatment with M3. (**b**) In the histogram, the results are expressed as mean fluorescence intensity in AU565. (**) *p* ≤ 0.010; (***) *p* < 0.001. Data are shown as mean ± SD of at least two independent experiments performed in technical duplicates (n = 4).

**Table 1 antioxidants-12-01560-t001:** Samples collected at different extraction times (min), temperatures (°C), and specific energy (Wh per kg of fresh raw material).

Fractions	Process Time (Min)	Process Temperature (°C)	Specific Energy (Wh/kg)
M1	5	24.0	13
M2	10	27.5	25
M3	15	30.5	38
M4	25	36.5	62
M5	45	47	108

**Table 2 antioxidants-12-01560-t002:** Cytotoxic effect of M3 extract treatments on AU565 and PBMCs from healthy donors (n = 3). Mean ± SD of three independent measurements of Trypan blue assay.

Cells Type	EC30 (mg/mL)	LD30 (mg/mL)
AU565	2.44 ± 0.01	>50
PBMCs	>50	>50

**Table 3 antioxidants-12-01560-t003:** Percentage of annexin V (ANX-V) and propidium iodide (PI) negative and positive AU565 cells treated with M3 extract for 4 h. Results are reported in percentage as mean ± SD of two independent experiments. (*) *p* ≤ 0.050; (**) *p* ≤ 0.010.

**AU565**	**ANX+/7AAD+ (%)**	**ANX+/7AAD− (%)**
CTR	1.28 ± 0.30	4.74 ± 0.25
H_2_O	0.94 ± 0.02	3.66 ± 0.18
ETO 50 µM	6.16 ± 0.10	3.86 ± 0.28
2.5 mg/mL	1.5 ± 0.15	4.98 ± 0.01
3.75 mg/mL	2.47 ± 0.25	9.74 ± 0.7
5 mg/mL	2.92 ± 0.23	14.28 ± 1.14 *
25 mg/mL	14.15 ± 1.35 **	17.6 ± 2.56 **
**PBMCs**	**ANX-V+/7AAD+ (%)**	**ANX-V+/7AAD− (%)**
CTR	4.73 ± 0.30	0.84 ± 0.15
H_2_O	0.92 ± 0.10	0.75 ± 0.35
ETO 50 µM	0.81 ± 0.36	3.42 ± 0.25
2.5 mg/mL	2.45 ± 0.45	0.65 ± 0.28
3.75 mg/mL	3.16 ± 0.12	0.79 ± 0.14
5 mg/mL	9.03 ± 0.18	1.24 ± 0.32
25 mg/mL	4.72 ± 0.45	4.06 ± 0.14

## Data Availability

The data presented in this study are available on request from the corresponding author.

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
