# Peer review of "Antioxidant Phytocomplexes Extracted from Pomegranate (*Punica granatum* L.) Using Hydrodynamic Cavitation Show Potential Anticancer Activity In Vitro"

_antioxidants, 2023, doi:10.3390/antiox12081560_

Round 1

Reviewer 1 Report

Antonella Minutolo et al investigate the biochemical properties and antioxidant power of collected fractions (from hydrodynamic cavitation) of pomegranate. They further test one of the extracts in a human cancer cell line and PBMC and reveal anti cancer proliferation and ROS scavenging properties. Although the topic of the manuscript is interesting and could provide substantial novel information to many scientific fields, many clarifications and expansion of validation methods for auhtors’ model is needed.

1.     It is not clear what authors refer to when using the collected fractions “as such”, please specify.

2.     At the end of the abstract is unclear if authors mean that the mechanism of anticancer effect relies on reduction of oxidative stress.

3.     The broad introduction on cancer could be much shortened.

4.     It would be important that authors highlight the novelty and importance of their study compared to others already done on pomegranate extracts the authors themselves cite. Is the HC method different than what has been applied before?

5.     A major concern is that authors only use 1 cancer cell line and 3 healthy donors as source of PBMCs. It would be mandatory that authors validate the results on cancer lines on at least 2 more human breast cancer lines (maybe from different cancer types like triple negative) and more donors as source of PBMCs (at least n=10).

6.     Authors should explain the rationale of investigating role of the pomegranate extract on PBMCs. Moreover, they should speculate and provide reference regarding why their compounds do not show substantial effect in PBMC vs cancer cells.

7.     A major explanation is needed for condensed tannins. Authors explain a bit in the discussion but what they are and why they are important should be included when they are mentioned in the results. A similar comment applies to anthocyanins and cyanidin.

8.     Authors should draw a clear conclusion from fig 1 and organically reason the major difference observed for A. On the same note, a better conclusion show be explained for table S1.

9.     It would be important to apply statistical test for fig 2.

10.  Authors mention that rosmarinic acid and myricetin have “low” absolute values. However it is unclear compared to what these are considered low.

11.  It would be mandatory to explain in the main text briefly how ABTS and FRAP test are performed: figure 3 indeed is not clear and hard to understand if information on the parameters assessed is not given.

12.  Please correct the typo in figure 3 which currently is named figure 2.

13.  Authors should provide more reasoning as of why they choose M3 for additional experiments.

14.  Etoposide treatment should be explained: is it used as a positive control? Please provide references in support.

15.  Please provide bar graphs with individual points (scatter plots) for all figures.

16.  Please indicate if anything is significant in figure 4a.

17.  Authors should assess viability also with MTT.

18.  It is not clear how many technical, “biological” replicates where used for the cell line viability assays.

19.  Authors should be more clear on reasoning for assessing viability as lack of proliferation vs cell death.

20.  Authors should speculate regarding the findings that at dose 3.75mg/ml authors already detect significant decrease of proliferation reflected by viability assay, while significant apoptosis is described only at 25mg/ml dose.

21.  It would be important to explain why ETO treatment does not elicit significant changes in apoptosis.

22.  Authors should explain better and clearly the conditions and timing after treatments with which viability/ apoptosis / ROS assays were performed.

23.  It is not clear what advantage table 3 provides that is not already included in fig 6. Please explain this aspect as table 3 appears redundant.

24.  It would be important to explain if the cancer cells were treated in any way (serum starvation/ cytokines etc) before assessing ROS. Do these cells just left untreated are ROS producers? Please provide references for underlining mechanisms. On this aspect, authors could stimulate PBMC in a way so they also would produce ROS, and then assess the antioxidant role of the extracts.

25.  About tannins, authors should better explain why they choose not to assess fraction A: is this related to the fact that condensed tannins are not stable in the purification process?

26.  It is not clear why authors did not choose to test M4 as well, considering its content in chlorogenic acid and quercetine 3-O glucoside. It would be important that authors included M4 in their experiments in cancer cells.

27.  When authors mention reported effects of anti cancer properties of pomegranate extracts, they should briefly describe the molecular mechanism behind those effects.

n/a

Reviewer 2 Report

The manuscript entitled “Antioxidant Phytocomplexes Isolated From Pomegranate (Punica granatum L.) Using Hydrodynamic Cavitation Reveal Potential Application as Adjuvants in Cancer Therapies investigated the extraction of phytochemicals from pomegranate (Punica granatum L.) fruits using HC, as well as the evaluation of  biochemical profile and in vitro antioxidant power of several fractions collected during the process. The antiproliferative effect of L-M3 was tested on human breast cancer line (AU565-PAR) and peripheral blood mononuclear (PBMCs) cells from healthy donors.

The manuscript is very interesting and well-written. However, minor revisions should be made in order to be published in Antioxidants journal, and the manuscript should be completed and/or modified taking into account the suggestions below:

1.      The authors are advised to rephrase the sentences from lines 50-52, 77-82, 95-98, 108-110, 271-273, 336-337, 487-489.

2.      The authors are advised to also provide the values with the content of metabolites in a table, too, for better understanding (From fig. 1 and 2).

3.       The authors are advised to note (in the figure legends Fig. 1 and 2) how many determinations were done for each sample?

4.      The authors are advised to complete the Discussion section, as the obtained results should be compared with those from other researches

Minor editing of English language required

Reviewer 3 Report

The manuscript entitled “Antioxidant Phytocomplexes Isolated from Pomegranate (Punica granatum L.) Using Hydrodynamic Cavitation Reveal Potential Application as Adjuvants in Cancer Therapies” reported the potential use of HC for exploiting antioxidants from P. granatum which are probably applied as pharmaceutical additives for cancer treatments.

 The manuscript is well-structuralized and the results are significant. This is typical for a study on natural products. However, to be published by Antioxidants, authors should seriously consider the following points:

1. The active antioxidant complexes obtained from pomegranate fruits by HC were not exactly isolated by a specific technique such as column chromatography. Hence, please consider changing the word “isolate” to another suitable word. In addition, the term “adjuvants” may be not proper in the case of the current results. Please change.

2. The basis of using different extraction times, temperatures, and specific energy in HC should be mentioned. From that, the optimal condition for extracting the highest bioactive compounds should be highlighted in the abstract but not only in conclusions.

3. Why the authors did not measure the total anthocyanins? Besides, the quantification of specific anthocyanins by using the mentioned spectrometric method is costly (time, labor, and fee). Why the authors did not use the HPLC-DAD method which is more advantageous?

4. The identification and specimen voucher of P. granatum sample should be provided.

5. A detailed discussion on advantages of HC method compared to other extraction methods for obtaining antioxidant complexes should be added.

6. A high similarity was detected by using iThenticate software. The authors should consider to paraphrase the following sections: L56-65; L86-92; section 2.7; section 2.8; section 2.10; L460-466.

To sum up, I suggest a major revision for this manuscript.

The manuscript is well-written. Some minor typos should be revised.

Round 2

Reviewer 3 Report

The authors have addressed the majority of my comments; however, there are still two points that need to be considered, as outlined below:

In point 3, I do not agree with the viewpoint that total anthocyanins can be determined by the aluminum chloride assay. The author should understand the reaction mechanism of each reagent with the samples extracted from natural sources. Aluminum chloride has a high affinity only for the hydroxyl (OH) groups of flavonols compounds or flavonol glycosides. On the other hand, anthocyanins have a different chemical structure and are not typically measured using the aluminum chloride assay. There are specific methods available for determining total anthocyanin content, such as pH differential or spectrophotometric methods using pH-dependent color changes.

Besides, there are several reasons resulted in the insignificance in HPLC-DAD results of the authors. Selection of a suitable column can be the first reason. The second is the mobile solvents.

In point 4, I agree with the presented identification, however, the specimen voucher and the place where it is deposited must be provided. The authors should remember that this is one of the essential points in studies on the application of natural compounds in the fields of medicine and therapeutics.

Overall, I suggest a minor revision for this manuscript.

 Minor editing of English language required
